# Salivary Metabolomic Analysis Reveals Amino Acid Metabolism Shift in SARS-CoV-2 Virus Activity and Post-Infection Condition

**DOI:** 10.3390/metabo13020263

**Published:** 2023-02-11

**Authors:** Tatiana Kelly da Silva Fidalgo, Liana Bastos Freitas-Fernandes, Barbara Bruno Fagundes Marques, Caroline Souza de Araújo, Bruno Jefferson da Silva, Taísa Coelho Guimarães, Ricardo Guimarães Fischer, Eduardo Muniz Barretto Tinoco, Ana Paula Valente

**Affiliations:** 1Department of Preventive and Community Dentistry, School of Dentistry, Universidade do Estado do Rio de Janeiro, Rio de Janeiro 20551-030, Brazil; 2National Center for Nuclear Magnetic Resonance, Medical Biochemistry, Universidade Federal do Rio de Janeiro, Rio de Janeiro 21941-902, Brazil; 3Department of Periodontology, School of Dentistry, Universidade do Estado do Rio de Janeiro, Rio de Janeiro 20551-030, Brazil

**Keywords:** saliva, COVID-19, SARS-CoV-2, metabolomics, post-COVID-19, PCR, nuclear magnetic resonance

## Abstract

The SARS-CoV-2 virus primarily infects salivary glands suggesting a change in the saliva metabolite profile; this shift may be used as a monitoring instrument during SARS-CoV-2 infection. The present study aims to determine the salivary metabolomic profile of patients with and post-SARS-CoV-19 infection. Patients were without (PCR−), with SARS-CoV-2 (PCR+), or post-SARS-CoV-2 infection. Unstimulated whole saliva was collected, and the ^1^H spectra were acquired in a 500 MHz Bruker nuclear magnetic resonance spectrometer at 25 °C. They were subjected to multivariate analysis using principal component analysis (PCA) and partial least squares discriminant analysis (PLS-DA), as well as univariate analysis through *t*-tests (SPSS 20.0, IL, USA), with a significance level of *p* < 0.05. A distinction was found when comparing PCR− subjects to those with SARS-CoV-2 infection. When comparing the three groups, the PLS-DA cross-validation presented satisfactory accuracy (ACC = 0.69, R2 = 0.39, Q2 = 0.08). Seventeen metabolites were found in different proportions among the groups. The results suggested the downregulation of major amino acid levels, such as alanine, glutamine, histidine, leucine, lysine, phenylalanine, and proline in the PCR+ group compared to the PCR− ones. In addition, acetate, valerate, and capronic acid were higher in PCR− patients than in PCR+. Sucrose and butyrate were higher in post-SARS-CoV-2 infection compared to PCR−. In general, a reduction in amino acids was observed in subjects with and post-SARS-CoV-2 disease. The salivary metabolomic strategy NMR-based was able to differentiate between non-infected individuals and those with acute and post-SARS-CoV-19 infection.

## 1. Introduction

Around the world, COVID-19 has affected many individuals and entire families. Among the countries currently most affected by COVID-19 worldwide, Brazil takes the third position in the ranking of the case–fatality ratio and second in the deaths per 100,000 population [1]. The latest numbers demonstrate that Brazil has recorded 34,696,863 COVID-19 cases and 685,978 deaths up to October 2022 [2]. The country presented 2.7% of the global population and accounted for 9.7% of infected and 12.6% of deaths worldwide by April 2021 [3]. Many factors can explain these numbers, one of which is the fake news industry financed by the actual government; among many disastrous actions, the purchase of ineffective medicines instead of vaccines and the payment of social influencers to advocate early treatment with hydroxychloroquine and ivermectin are highlighted [4]. The delayed acquisition of vaccines and other unsuitable actions turned Brazil into a global epicenter in the pandemic in 2020–2021 [5]. In addition, specific personal responses could increase the infectivity of SARS-CoV-2 in the Brazilian population.

Saliva is an informative biofluid that presents blood components such as proteins and low molecular weight metabolites. This exchange occurs in the thin layer of epithelial cells separating the salivary ducts from the systemic circulation, allowing the transfer of these metabolites to the saliva through active and passive transport and diffusion through the membrane. In addition, gingival crevicular fluid, an inflammatory exudate between gingival tissue and tooth, also carries blood components [6]. The salivary balance is crucial to maintaining oral homeostasis and determining physiological and pathological conditions [7]; the salivary profile may eventually modify after SARS-CoV-2 infection.

The SARS-CoV-2 spike can bind to ACE2 (hACE2) human receptor through its receptor-binding domain (RBD) and TMPRSS family members, which are expressed in epithelial cells of the oral mucosa [8,9] and salivary gland [8,9,10,11]. Thus, this mechanism suggests that SARS-CoV-2 infection can result in a local modification of the metabolism of salivary glands and result in changes in the salivary metabolites profile. Furthermore, the salivary metabolic fingerprint alterations may reflect the gland cell metabolism [12]. 

Several efforts are being carried out to understand the course of COVID-19 at the level of the metabolite phenotype since the changes are related to the patient’s physical condition and result from modifications of the local microenvironment while affecting cellular metabolism [13,14,15]. Previous studies have demonstrated alterations in the blood metabolome profile of patients with COVID-19 [14,15]. The authors found altered metabolism related to carnitines, ketone bodies, fatty acids, lysophosphatidylcholines/phosphatidylcholines, tryptophan, bile acids, and purines. It was also pointed out that phenylalanine plays a role in COVID-19 as a disease severity marker [16]. Furthermore, alterations in the levels of ketone bodies and hydroxybutyric acid, as well as a reduction in amino acid and kynurenine metabolic pathways during SARS-CoV-2 infection, influence the systemic severity of the disease [17,18]. 

Salivary glands are primarily infected, so an impact of disease in saliva metabolism is expected; changes to the salivary metabolites may be used as a monitoring instrument during SARS-CoV-2 infection [11]. In this sense, the metabolomic NMR-based approach applied to saliva can identify hundreds of endogenous and exogenous biomolecules, including glandular metabolism known as salivary secretion and gingival fluid, which transports systemic metabolites, dietary compounds, oral health products, and pharmaceuticals [19]. Our group showed salivary metabolites’ shift in different local and systemic disturbances using the metabolomics NMR-based strategy, including type I diabetes [20], renal failure [21,22], and dental caries [23,24]. To better understand the phases of SARS-CoV-2 infection, this study aimed to characterize the metabolomic changes during the SARS-CoV-2 infection phase and post-disease condition.

## 2. Experimental Design

### 2.1. Subjects and Saliva Collection

A total of thirty-five patients over 18 years old from Piquet Carneiro Polyclinic at State University of Rio de Janeiro (PPC—UERJ) were recruited during an RT-PCR screening for SARS-CoV-2, being separated into those without SARS-CoV-2 (PCR−; n = 9) and with SARS-CoV-2 infection (PCR+; n = 8) groups. The third group was composed of hospitalized subjects due to COVID-19 complications (post-SARS-CoV-2 infection; n = 18). This group was previously diagnosed with SARS-CoV-2 infection, confirmed by RT-PCR testing, but at the time of sample collection, the subjects were submitted to another RT-PCR and received a negative diagnostic result, so they were classified as post-SARS-CoV-2 infection. Therefore, the patients were grouped into a negative RT-PCR group without SARS-CoV-2 infection (PCR−) and the positive RT-PCR (PCR+) and post-SARS-CoV-2 infection groups. The characteristics of subjects, signals, and symptoms of included subjects were self-reported and were obtained through a questionnaire.

Saliva was collected under a complete biosafety protocol (coat, face shield, goggles, N-95, gloves changed at each collection, surgical clothes, and closed-toe shoes), including the disposal of examination and collection material. To be included in the study, subjects must refrain from oral activities for one hour before saliva collection. For the collection of saliva samples, participants were required to expectorate 3 mL of unstimulated whole saliva into a universal plastic tube that was kept on ice until centrifugation. In addition, Triton X-100 (Sigma-Aldrich, USA) in a concentration of 0.25% *v*/*v* was added to a saliva tube to avoid viral contamination. Saliva samples were centrifuged at 10,000× *g* for 60 min at 4 °C, and supernatants were stored at −80 °C until NMR analyses.

### 2.2. Sample Preparation and NMR Measurements

NMR spectra were acquired using a 500-MHz Avance III spectrometer (Bruker Biospin, Rheinstetten, Germany). Saliva samples were prepared by mixing 0.54 mL of salivary supernatant, 0.06 mL of phosphate buffer pH 7.0 containing D_2_O (99.8%; to provide a field frequency lock), and 20 µM of 4,4-dimethyl-4-silapentane-1-sulfonic acid (DSS) for chemical shift referencing, δ = 0.00 ppm. The ZGPR pulse was used for saliva samples at 298 K with 1024 scans. In addition, ^1^H-^1^H total correlation spectroscopy (TOCSY) experiments were conducted with acquisition parameters of 4096 × 256 complex points and a mixing time of 70 ms.

After spectra acquisition, edge effects were evaluated by overlaying all spectra using Topspin (Bruker Biospin). The spectra and spectral regions that could not be corrected for phase and baseline were excluded from the analyses. In the present study, all spectra presented good quality; therefore, all thirty-five spectra were included. Our assignment strategy included using the Human Metabolome database (http://www.hmdb.ca/ accessed on 18 August 2022) and previous studies [19,20,21,23,24]. TOCSY experiments were used to confirm and assign metabolites.

### 2.3. Statistical Analysis

Metabolomic data were extracted using the AMIX statistical program (Bruker Biospin). Each NMR spectrum was analyzed by integrating bucket sizes of 0.03 ppm without the water region (4.5–5.5 ppm) considering 0.5–8.5 ppm. The respective area of the NMR spectrum, which corresponds to the Triton X-100 peaks (0.55–0.82, 1.23–1.30, 1.58–1.72, 3.49–3.79, 3.96–4.09, 6.71–6.89, 7.12–7.26 ppm), was removed. The total number of buckets and, therefore, possible metabolites was 225. After the multivariate and univariate analysis, the buckets that presented statistical differences were assigned as the correspondent metabolites. 

The saliva spectra datasets were stored in a matrix, with rows representing the subject samples and columns representing chemical shifts. The bucket table was normalized using the sum of signal intensities and subjected to the Pareto scaling method [25] before applying the principal component analysis (PCA) and partial least squares discriminant analysis (PLS-DA) method. The PLS-DA used the initial input variables for each group stored in the y-table (0 and 1) and was performed using Metaboanalyst 5.0 software (www.metaboanalyst.ca/MetaboAnalysts accessed on 12 August 2022) [26]. 

Metaboanalyst 3.0 software was also used to obtain the predictive performance of the models; each model was evaluated for Q^2^, R^2^, and accuracy (ACC) for cross-validation [27]. A 95% confidence interval was used for the models.

The variable importance in projection (VIP) scores were obtained for each comparison, and the corresponding metabolites were analyzed using peak intensity and univariate statistical analyses (*p* < 0.05). All spectral regions (buckets) were subjected to univariate statistical analyses for further analysis. NMR data distribution was also assessed using the Shapiro–Wilk test (*p* < 0.05), demonstrating normal data, and the ANOVA test with Bonferroni correction was performed (*p* < 0.05). 

## 3. Results

Thirty-five patients were recruited after applying the inclusion and exclusion criteria. In the PCR− group (n = 9), five were female, with a mean age of 47.83 years (±22.03); in the PCR+ group (n = 8), three were female, with a mean age of 47.83 years (±22.03); and in the post-SARS-CoV-2 infection group (n = 18), seven were female, with a mean age of 63.61 years (±9.44). Table 1 shows that among the control group, none were smokers, lost their smell and taste, or presented with a fever. In the PCR+ group, 37.5% (n = 3) presented fever, 37.5% (n = 3) presented a cough, 62.5% (n = 5) reported malaise symptoms, and 25.0% (n = 2) lost their sense of smell and taste. In the post-SARS-CoV-2 infection group, 44.44% (n = 8) presented fever, 100% (n = 18) presented with a cough, 83.33 % (n = 15) indicated malaise symptoms, and 55.55% (n = 10) reported a loss of smell and taste.

The ^1^H-NMR spectra from PCR−, PCR+, and post-SARS-CoV-2 infection groups were obtained and assigned. Multivariate analyses were performed to evaluate the metabolite differences between the groups. Figure 1 shows the three groups and the comparisons between them. PCA demonstrated a slight separation between groups (Figure 1A) which became more evident with PLS-DA analysis (Figure 1B). The PLS-DA cross-validation presented satisfactory accuracy considering the first principal component (ACC = 0.69, R^2^ = 0.39, Q^2^ = 0.08). It is possible to observe that PCR− group presented different salivary metabolite profiles compared to PCR+ and post-SARS-CoV-2 infection. This difference is more evident when comparing PCR− and PCR+ subjects (Figure 1B)—demonstrating that after infection, the post-SARS-CoV-2 group presented a slight recovery. 

Appendix A shows salivary metabolites, chemical shifts, peak intensity variations, and *p*-values resulting from multivariate and univariate analyses. Figure 2 shows the metabolites that were highlighted in the VIP score, demonstrating an important variation within groups, especially in PCR− group, and in PCR+ was observed more outliers. These data indicate seventeen different assigned chemical shifts with a statistical difference of at least between two groups. Amino acids such as glutamine, proline, and lysine presented a downregulation in the PCR+ and post-SARS-CoV-2 infection subjects. Acetate, ethanol, and capronic acid also presented a reduction in PCR+ and post-SARS-CoV-2 infection subjects. Figure 3 exhibits the amino acids (histidine, leucine, and phenylalanine) that presented statistical differences (*p* < 0.05) in the univariate analysis. 

The main metabolic pathways affected (Figure 4) are glycolysis/gluconeogenesis (match status 4/26; *p* < 0.001), aminoacyl-tRNA biosynthesis (match status 3/48; *p* = 0.003), pyruvate metabolism (match status 2/22; *p* = 0.009), lysine degradation (match status 2/25; *p* = 0.012), glyoxylate and dicarboxylate metabolism (match status 2/32; *p* = 0.020), glutamine and glutamate metabolism (match status 1/6; *p* < 0.042) and nitrogen metabolism (match status 1/6; *p* = 0.042).

## 4. Discussion

COVID-19 is a systemic infection that causes a significant impact on metabolism, especially the downregulation of amino acid levels. This shift in metabolism was already observed by NMR alterations in plasma metabolites during SARS-CoV-2 infection and post-infection [17,18]. Li et al. (2022) also observed changes in plasma proteomics and the metabolomics of COVID-19 survivors six months after discharge from the hospital. The COVID-19 survivors exhibited significant differences in the extracellular matrix, immune response, and hemostasis pathways compared to the PCR− subjects [28]. The present study demonstrated that saliva could distinguish the presence and status of the SARS-CoV-19 infection into SARS-CoV-2 activity and post-SARS-CoV-2 infection condition. The role of amino acids in immune responses by regulating includes the activation of T lymphocytes, B lymphocytes, natural killer cells, and macrophages; cellular redox state, gene expression, and lymphocyte proliferation; and the production of antibodies, cytokines, and other cytotoxic substances [29]. The amino acids can exert regulatory effects at various levels of cell activity, acting as mediators or signal molecules and, therefore, modulating numerous functions, allowing the adequate expression of these regulatory activities in vivo during inflammatory states [30]. In addition, a correlation between metabolite changes and concentrations of inflammatory cytokines and chemokines points to an immunometabolic disorder [18]. Passlack et al. (2016) demonstrated that arginine, lysine, leucine, and threonine significantly impacted cytokine secretion and proliferative activity of the T cells [31]. Therefore, these findings suggest that amino acids shift can affect the immune response and influence the disease’s course.

The present study suggested a salivary downregulation of major amino acid levels, such as alanine, glutamine, histidine, leucine, lysine, phenylalanine, and proline in the SARS-CoV-2 infection and post-infection groups. It is important to highlight the variation within groups, especially in PCR− group. This fact can occur due to the interindividual expected variability. In addition, it is possible to observe that in the PCR+ group, more outliers can be observed, probably due to variability in individual response to SARS-CoV-2 infection. The findings demonstrated that saliva presented a similar metabolism shift in plasma [32,33]. Although the dysregulation in COVID-19 patients still needs elucidation, evidence suggests that these amino acid pathways are altered due to the inflammatory response [34] mediated by biochemical and metabolic pathways related to this disease [35,36]. The change in amino acid rates plays a role in the development of several diseases; metabolomic studies in recent years have shown significant differences in these levels. Low levels of circulating amino acids are a hallmark of coronavirus infections, including SARS-CoV-1, MERS-CoV, and Ebola, denoting a widespread pattern of the body’s reaction to viral infection [37,38,39,40]. The reprogramming of amino acids, glucose, and fatty acids is a peculiar feature of COVID-19 infection [32].

In our study, the metabolites that changed in saliva are related to histidine, phenylalanine, and taurine metabolism. A higher level of histidine was observed in the post-SARS-CoV-19 infection group compared to PCR+e individuals, which is an indication of recovery of the level of healthy saliva. In addition, histidine, a non-essential amino acid, is related to a protective effect against SARS-CoV-2 infection [41]. The histidine metabolism pathway is also related to pos-SARS-CoV-19 infection and PCR− subjects. Although this finding may reflect the recovery process along the post-disease stage, disease severity also affected the pathways. In inflammatory settings such as trauma, sepsis, and critical illness, the relevance of taurine is emphasized as the body systems have reduced anti-inflammatory protection.

In the present study, phenylalanine was decreased in PCR+ subjects compared to PCR−. The literature reported that phenylalanine was significantly lower in blood in mild disease compared to moderate and severe groups, which could be a marker of disease severity [16]. The finding suggests that the metabolite changes, especially amino acids, correlate with the concentrations of inflammatory cytokines and chemokines, resulting in an immune-metabolic disorder [16,18,42]. Huang et al. observed that phenylalanine- and leucine-defined metabolic types show a high mortality risk in patients with severe infection. They found that an increase in phenylalanine and leucine levels in the blood is related to the severity of COVID-19 infection. We also found different levels of phenylalanine and leucine in saliva from PCR− subjects in comparison with subjects with SARS-CoV-2 infection and post-infection. Still, we also saw a decrease in the levels. It is important to note that phenylalanine appears to be a severity marker related to respiratory diseases, such as acute respiratory distress syndrome (ARDS), when compared with PCR− subjects [8]. Therefore, it is reasonable to investigate phenylalanine in saliva in patients post-COVID-19 further and correlate it with the severity of clinical parameters. The literature reports that phenylalanine metabolism was one of the important metabolic pathways affected by the SARS-CoV-2 virus [43] which aligns with our findings. NMR plasma metabolic analysis has shown metabolite changes when comparing PCR− subjects to PCR+ individuals, including a reduction in some essential amino acids [17]. Detecting systemic biomarkers related to multiple systems gives the potential for understanding systemic recovery and severe cases detected in saliva.

A reduction of alanine in PCR+ and post-SARS-CoV-2 subjects was observed, corroborated with previous findings [32]. The catabolism of glutamate is impacted in subjects with COVID-19 [32], and glutamate is a building block for the synthesis of a variety of amino acids, including alanine and proline, that was reduced in the present study. ShihokoKimura-Ohba et al. (2023) suggested D-Alanine as a COVID-19 and severe influenza virus infection biomarker and indicated the supplementation with D-Alanine to alleviate the severity of the clinical course [44].

The present study demonstrated acetate levels in a higher proportion in positive PCR individuals compared to the post-SARS-CoV-19 infection group. In the COVID-19 context, the lower blood pH presents a worse prognostic factor for death and critical illness [41]. Our previous study has demonstrated a relationship between oral conditions and organic acids in the saliva, such as acetic and capronic, probably produced by oral microorganisms [23,24,45].

Glucose metabolism is affected by COVID-19 at various stages during the disease course [46]. We found lower glucose levels and sugars during SARS-CoV-2 infection than in healthy subjects or those post-SARS-CoV-19 infections, with the opposite being reported for sucrose levels. Thomas et al. (2020) found an increase in blood glucose levels [33].

The limitations of the present study include common confounders such as differences in age, sex, and BMI. The molecular mechanisms that define host−virus interactions in SARS-CoV-19 infection are critical to both the clinical approach and diagnostic tools that can indicate patients into risk categories. Identifying complex metabolic and systemic relationships associated with the host response to viral infection may be described as a specific pathway directly related to acute infection. The saliva metabolism of patients with COVID-19 changed specifically and reflected the influence of the disease stages. Therefore, a better understanding of the disease, identifying altered metabolic pathways and candidates’ metabolites characteristic of the disease and disease stages (PCR+ and SARS-CoV-19 infection) can facilitate personalized treatment, monitoring, and disease control. It is suggested that this knowledge can lead to better-personalized medicine strategies for this disease core, such as early diagnostic and metabolites supplementation, preventing worse outcomes.

## 5. Conclusions

The present study demonstrated that non-infected (PCR−), infected (PCR+), and post-SARS-CoV-19 infection individuals presented different salivary fingerprints, especially a disease-related downregulation of amino acid levels. Therefore, the metabolites found might help to understand the status of SARS-CoV-19 infection and monitor its effects.

## Figures and Tables

**Figure 1 metabolites-13-00263-f001:**
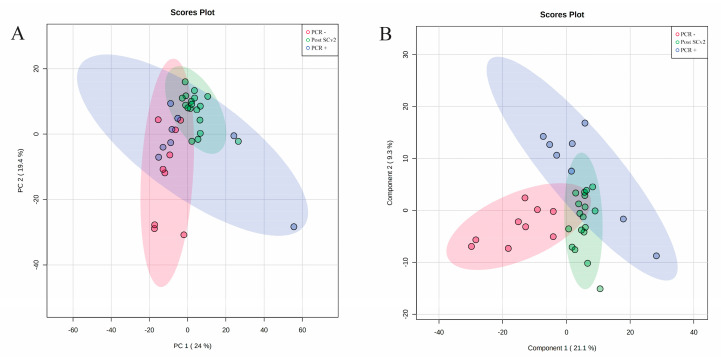
(**A**) PCA; and (**B**) PLS-DA comparing subjects from PCR−, PCR+, and post-SARS-CoV-2 infection (post-SCv2).

**Figure 2 metabolites-13-00263-f002:**
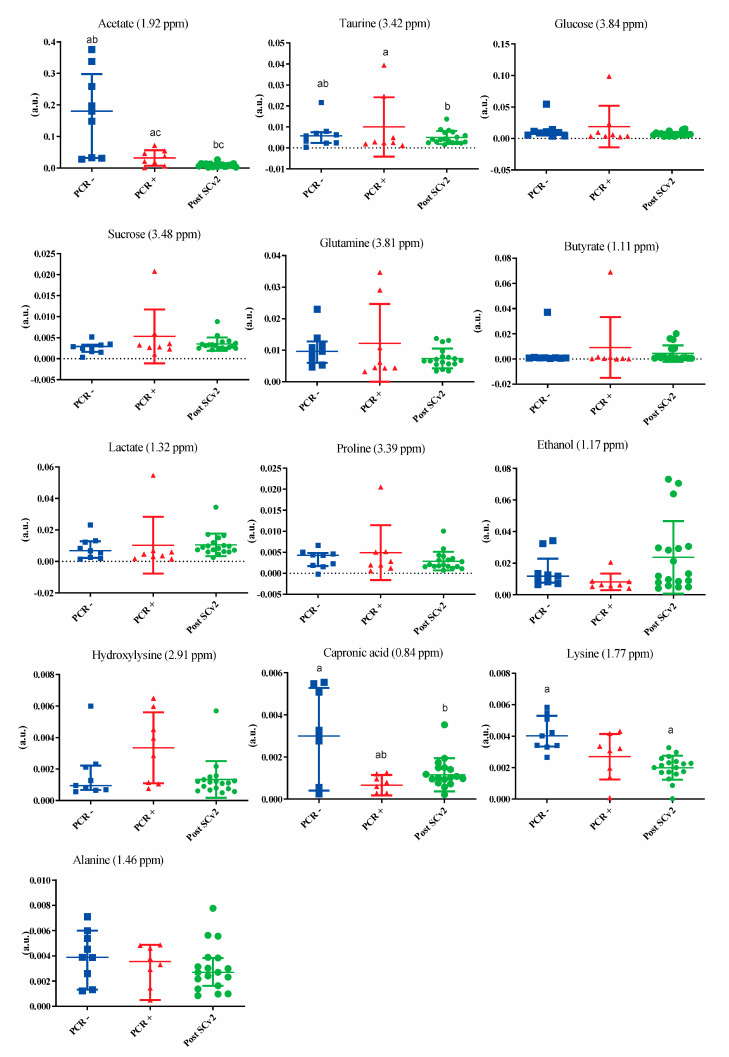
Metabolites highlighted by VIP score demonstrate the differences between PCR−, PCR+, and post-SARS-CoV-2 infection (post-SCv2). The metabolites that presented statistical differences in the univariate analysis (Appendix A) are indicated using letters; the same letter indicates a statistical difference.

**Figure 3 metabolites-13-00263-f003:**
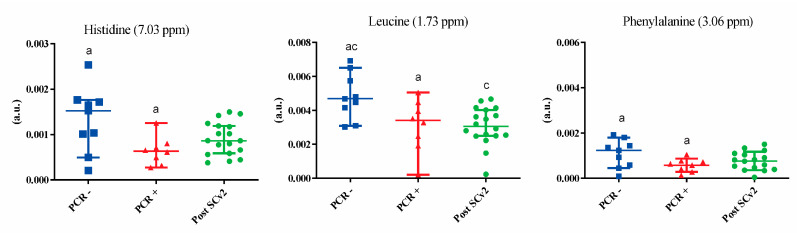
Amino acids that presented statistical differences (*p* < 0.05) in the univariate analysis for the groups PCR−, PCR+, and post-SARS-CoV-2 infection (post-SCv2). The metabolites that presented statistical differences in the univariate analysis (Appendix A) are indicated using letters; the same letter indicates a statistical difference.

**Figure 4 metabolites-13-00263-f004:**
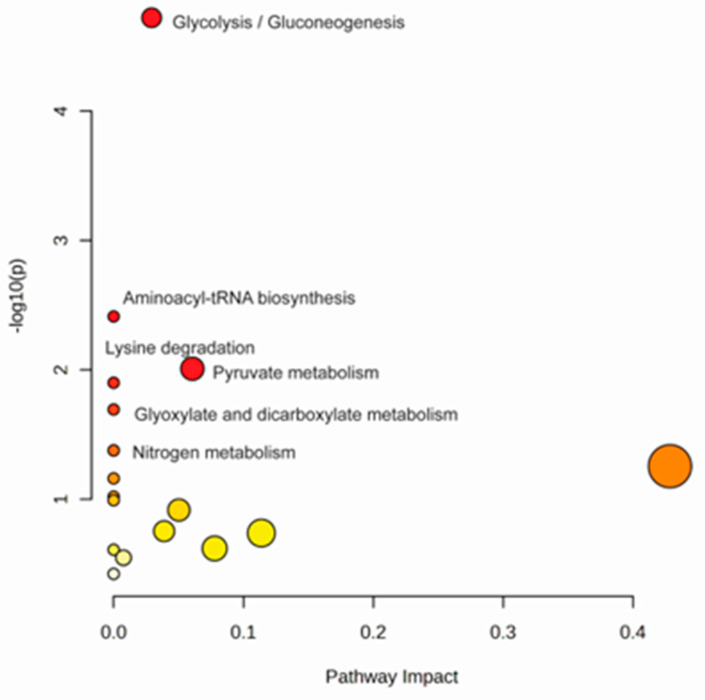
Metabolite pathways related to SARS-CoV-2 infection, considering the metabolites of VIP score.

**Table 1 metabolites-13-00263-t001:** Characteristics, signals, and symptoms of included subjects.

**Signals/Symptoms**	**PCR−**	**PCR+**	**Post-S** **ARS-** **CoV-2 Infection**
**Smokers** **Ex-smokers**	0% (n = 0) 0% (n = 0)	0% (n = 0) 0% (n = 0)	0% (n = 0) 44.44 (n = 8)
**Lost smell or taste**	0% (n = 0)	25.0% (n = 2)	55.55% (n = 10)
**Fever**	0% (n = 0)	37.5% (n= 3)	44.44 (n = 8)
**Caught**	22.22 (n = 2)	37.5% (n = 3)	100% (n = 18)
**Malaise symptom**	33.33% (n = 3)	62.5% (n = 5)	83.33 % (n = 15)

## Data Availability

The data are available at https://doi.org/10.17605/OSF.IO/VFXRA (accessed on 2 January 2023).

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
