# Peer review of "Salivary Metabolomic Analysis Reveals Amino Acid Metabolism Shift in SARS-CoV-2 Virus Activity and Post-Infection Condition"

_metabolites, 2023, doi:10.3390/metabo13020263_

Round 1

Reviewer 1 Report

The authors presented a NMR-based metabolomics approach to find the metabolic profile differences among three groups related to covid. It is an interesting to point to find potential biomarkers for the disease. However, several concerns need to be addressed.

Abstract needs a few sentences on the background and reason for performing this kind of study, and in the end, needs a sentence on the significance of the study/findings.

Line 40: what is post-SARS-CoV-2 group comparing to in terms of sucrose and butyrate? 

The number of patients recruited for each group should be mentioned in the method section.

Reference missing for Line 67-68 and 76-79.

More details are needed for the sample collection part, especially in consideration of inappropriate sample collection resulting in metabolites of food consumption.

In Line 127, "spectra and spectra regions that could not be corrected for phase... were excluded ..." Does this affect the final replicate number of each group? 

How many metabolites were identified for further data analysis? This should be included in the result section.

Figure 2 plots should have marks for significance.

Figure 3 is not mentioned in the main text.

Pathway analysis showed D-glutamine metabolism affected, but the majority of amino acids in human body should be L form not D form. 

Several gramma errors showed up in the manuscript, please revise.

Author Response

The authors presented a NMR-based metabolomics approach to find the metabolic profile differences among three groups related to covid. It is an interesting to point to find potential biomarkers for the disease. However, several concerns need to be addressed.

R: We acknowledge for the comment. In fact, the finding eligible biomarkers and the understanding of the diseases course using saliva as biofluid is an important knowledge filled lacune. We also hope that all changed helped to improve the manuscript’s quality.

Abstract needs a few sentences on the background and reason for performing this kind of study, and in the end, needs a sentence on the significance of the study/findings.

R: The abstract was changed, and a background sentence was included:

“The SARS-CoV-2 virus primarily infects salivary glands suggesting a change in the saliva me-tabolite profile; this shift may be used as a monitoring instrument during SARS-CoV-2 infection.”

Line 40: what is post-SARS-CoV-2 group comparing to in terms of sucrose and butyrate? 

R: The sentence was corrected to:

“Sucrose and butyrate were higher in post-SARS-CoV-2 infection compared to PCR-.”

The number of patients recruited for each group should be mentioned in the method section.

R: This information was included in the method session:

“A total of thirty-five patients over 18 years old from Piquet Carneiro Polyclinic at State University of Rio de Janeiro (PPC – UERJ) were recruited during an RT-PCR screening for SARS-CoV-2, being separated into those without SARS-CoV-2 (PCR-; n= 9) and with SARS-CoV-2 infection (PCR+; n= 8) groups. The third group was composed of hospitalized subjects due to COVID-19 complications (post-SARS-CoV-2 infection; n =18.”

Reference missing for Line 67-68 and 76-79.

R: The references were included:

“The salivary balance is crucial to maintaining oral homeostasis and determining physiological and pathological conditions [7]; the salivary profile may eventually modify after SARS-CoV-2 infection.”

“Several efforts are being carried out to understand the course of COVID-19 at the level of the metabolite phenotype since the changes are related to the physical condition of the patients and result from modifications of the local microenvironment while affecting cellular metabolism [13-15].”

More details are needed for the sample collection part, especially in consideration of inappropriate sample collection resulting in metabolites of food consumption.

R: The information related to sample collection and fasting were added:

“To be included in the study, subjects must refrain from oral activities for one hour before saliva collection. For the collection of saliva samples, participants were required to expectorate 3 mL of unstimulated whole saliva into a universal plastic tube that were kept on ice until centrifugation.”

In Line 127, "spectra and spectra regions that could not be corrected for phase... were excluded ..." Does this affect the final replicate number of each group?

R: In the present study, all spectra presented good quality, therefore it did not influence the final number of spectra:

“In the present study, all spectra presented good quality, therefore all thirty-five spectra were included.”

How many metabolites were identified for further data analysis? This should be included in the result section.

R: We used a strategy of integrating the bucket sizes of 0.03 ppm. Therefore, the spectra were divided into 225 buckets. These buckets were submitted to statistical analysis and the buckets that presented statistical difference were assigned as the correspondent metabolite. This strategy was included in the text to make clearer to readers:

 “Metabolomic data were extracted using the AMIX statistical program (Bruker Bio-spin). Each NMR spectrum was analyzed by integrating bucket sizes of 0.03 ppm without the water region (4.5–5.5 ppm) considering 0.5–8.5 ppm. (…) The total number of buckets and, therefore, possible metabolites were 225. After the multivaried and univaried analysis, the buckets that presented statistical differences were assigned as the correspondent metabolite.”  

Figure 2 plots should have marks for significance.

R: All metabolites assigned in VIP score (multivaried analysis) presented statistical difference, we complemented the analysis with univariated analysis. This information was included in the text and the marks for univaried analysis were added in the Figure 2. We also included this misinformation in Figure 3.

“The variable importance in projection (VIP) scores were obtained for each comparison, and the corresponding metabolites were analyzed using peak intensity and univariate statistical analyses (p < 0.05). All spectral regions (buckets) were subjected to univariate statistical analyses for further analysis. NMR data distribution was also assessed using the Shapiro-Wilk test (p<0.05), demonstrating normal data, and The ANOVA test with Bonferroni correction was performed (p<0.05).”

Figure 2 and 3 was modified and was included letters to indicate statistical difference.

Figure 2: Metabolites highlighted by VIP score demonstrating the differences between PCR-, PCR+, and post-SARS-CoV-2 infection (post-SCv2). The metabolites that presented statistical differences in the univariate analysis (Supplementary Figure 1) are indicated using letters; the same letter indicates a statistical difference.

 Figure 3: Amino acids that presented statistical differences (p<0.05) in the univariate analysis for the groups PCR-, PCR+ and post-SARS-CoV-2 infection (post-SCv2). The metabolites that presented statistical differences in the univariate analysis (Supplementary Figure 1) are indicated using letters; the same letter indicates a statistical difference.

  • Figure 3 is not mentioned in the main text.

R: The authors acknowledge for this appointment. The Figure 3 was reported in the text:

“Figure 3 exhibits the amino acids (histidine, leucine, and phenylalanine) that presented statistical differences (p<0.05) in the univariate analysis.”

  • Pathway analysis showed D-glutamine metabolism affected, but the majority of amino acids in human body should be L form not D form. 

R: The authors acknowledge for this comment. We corrected the text and removed the isometric form to kept it in accordance to Figure 4.

  • Several gramma errors showed up in the manuscript, please revise.

R: The manuscript was submitted to a professional English revision prior to submission to Metabolites. Even thought, it was revised again to correct specific errors.

Reviewer 2 Report

General Comments 

Reviewed is the manuscript “Salivary metabolomic analysis reveals amino acid metabolism  shift in SARS-CoV-2 virus activity and post infection condition” submitted by Tatiana Kelly da Silva Fidalgo, et. al. The author looked at differences in the salivary metabolomes of individuals with and without SARS-CoV-2 infection, as well as those who had recovered from SARS-CoV-2 infection. The researchers found that certain amino acid levels and certain metabolites were different between the three groups. In addition, NMR-based salivary metabolomics can differentiate between non-infected individuals and those with acute and post-SARS-CoV-2 infection. This paper demonstrates clear structure and organization, with a coherent flow of information and few typographical errors. It can be accepted for publication with minor revisions. The authors have effectively described their methods and provided relevant performance data in the field of ranking and selection.

Specific Comment:

·         It is highly recommended that that multiple testing adjustments (FDR) be applied to account for the large number of statistical tests being performed in this study.

·         In table it would be helpful to provide more information on the symptoms being reported, such as how they were assessed and over what time period.

·         Make sure the font size is consistent for the whole manuscript. For instance, in line 189, “If an experimenter chose to detect a difference of at least…”, the font size here is different from its surroundings.

·         Figure 1, it would be helpful to provide more context for the results shown in the plots, such as how they relate to the research question being investigated or the specific hypotheses being tested.

·         In figure 2, huge in group variance can be found in PCR- group and outliers can be seen in PCR+ group, this should be discussed in the manuscript.

·         It would be useful to discuss the potential implications of the findings for the diagnosis and treatment of SARS-CoV-2 infection or the development of personalized medicine strategies.

·         It is not clear how the changes in amino acid levels and other metabolites observed in the present study relate to the inflammatory response or disease severity. It would be helpful to provide more context for the results of the present study and how they relate to the research question being investigated.

Author Response

Reviewer 2

It is highly recommended that that multiple testing adjustments (FDR) be applied to account for the large number of statistical tests being performed in this study.

R: The authors acknowledge for this suggestion; the univariate data were submitted to ANOVA with Bonferroni correction. The modified Supplemental Table 1 was uploaded.

“NMR data distribution was also assessed using the Shapiro-Wilk test (p<0.05) demonstrating normal data, and the ANOVA test with Bonferroni correction was performed (p<0.05).”

In table it would be helpful to provide more information on the symptoms being reported, such as how they were assessed and over what time period.

R: The characteristics of the subjects, signals, and symptoms were self-reported and were obtained by a standard questionnaire from hospital. The time period information was not mentioned. To make this point clearer in the text, this information was included:

“The characteristics of subjects, signals, and symptoms of included subjects were self-reported and were obtained through a questionnaire.”

Make sure the font size is consistent for the whole manuscript. For instance, in line 189, “If an experimenter chose to detect a difference of at least…”, the font size here is different from its surroundings.

R: The font size of manuscript (from introduction to conclusion) was adjusted to 10 (style: Palatino Linotype).

Figure 1, it would be helpful to provide more context for the results shown in the plots, such as how they relate to the research question being investigated or the specific hypotheses being tested.

R: The description of Figure 1 was improved:

“It is possible to observe that PCR- group presented different salivary metabolites profile compared to PCR+ and post-SARS-CoV-2 infection. This difference is more evident when comparing PCR- and PCR+ subjects (Figure 1B) - demonstrating that after infection, the post-SARS-CoV-2 group presented a slight recovery.

In figure 2, huge in group variance can be found in PCR- group and outliers can be seen in PCR+ group, this should be discussed in the manuscript.

R: This finding was included in the results and discussion session:

Results:

“Figure 2 shows the metabolites that were highlighted in the VIP score, demonstrating an important variation within groups, especially in PCR- group and in PCR+ was observed more outliers.”

Discussion:

“The present study suggested a salivary down-regulation of major amino acid levels, such as alanine, glutamine, histidine, leucine, lysine, phenylalanine and proline in the SARS-COV-2 infection and post-infection groups. It is important to highlight the variation within groups, especially in PCR- group. This fact can occur due to the interindividual expected variability. In addition, it is possible to observe that in the PCR+ group, more outliers can be observed, probably due to variability in individual response to SARS-COV-2 infection.

It would be useful to discuss the potential implications of the findings for the diagnosis and treatment of SARS-CoV-2 infection or the development of personalized medicine strategies.

R: This information was included in the discussion session:

“Therefore, a better understanding of the disease, identifying altered metabolic pathways and candidates’ metabolites characteristic of the disease and disease stages (PCR+ and SARS-CoV-19 infection) can facilitate personalized treatment, monitoring, and disease control. It is suggested that this knowledge can lead to better-personalized medicine strategies for this disease core, such as early diagnostic and metabolites supplementation, preventing worse outcomes.  

It is not clear how the changes in amino acid levels and other metabolites observed in the present study relate to the inflammatory response or disease severity. It would be helpful to provide more context for the results of the present study and how they relate to the research question being investigated.

R: This discussion was included in the manuscript:

“The COVID-19 survivors exhibited significant differences in the extracellular matrix, immune response, and hemostasis pathways compared to the PCR- subjects [29]. The present study demonstrated that saliva could distinguish the presence and status of the SARS-CoV-19 infection into SARS-CoV-2 activity and post-SARS-CoV-2 infection condition. The role of amino acids in immune responses by regulating includes the activation of T lymphocytes, B lymphocytes, natural killer cells, and macrophages; cellular redox state, gene expression, and lymphocyte proliferation; and the production of antibodies, cytokines, and other cytotoxic substances [30]. The amino acids can exert regulatory effects at various levels of cell activity, acting as mediators or signal molecules and, therefore, modulating numerous functions, allowing the adequate expression of these regulatory activities in vivo during inflammatory states [31]. In addition, a correlation between metabolite changes and concentrations of inflammatory cytokines and chemokines points to an immunometabolic disorder [18]. Passlack et al. (2018) demonstrated that arginine, lysine, leucine, and threonine significantly impacted cytokine secretion and proliferative activity of the T cells [32]. Therefore, these findings suggest that amino acids shift can affect the immune response and influence the disease’s course.”

Round 2

Reviewer 1 Report

Thank you for the response.